# Evolution of an extreme hemoglobin phenotype contributed to the sub-Arctic specialization of extinct Steller's sea cows

Anthony V Signore[1], Phillip R Morrison[2], Colin J Brauner[2], Angela Fago[3], Roy E Weber[3], Kevin L Campbell[1]*

[1]Department of Biological Sciences, University of Manitoba, Winnipeg, Canada; [2]Department of Zoology, University of British Columbia, Vancouver, Canada; [3]Department of Biology, Aarhus University, Aarhus, Denmark

*For correspondence: kevin.campbell@umanitoba.ca

**Competing interest:** The authors declare that no competing interests exist.

**Abstract** The extinct Steller's sea cow (*Hydrodamalis gigas*; †1768) was a whale-sized marine mammal that manifested profound morphological specializations to exploit the harsh coastal climate of the North Pacific. Yet despite first-hand accounts of their biology, little is known regarding the physiological adjustments underlying their evolution to this environment. Here, the adult-expressed hemoglobin (Hb; $\alpha_2\beta/\delta_2$) of this sirenian is shown to harbor a fixed amino acid replacement at an otherwise invariant position ($\beta/\delta82Lys\rightarrow Asn$) that alters multiple aspects of Hb function. First, our functional characterization of recombinant sirenian Hb proteins demonstrates that the Hb-$O_2$ affinity of this sub-Arctic species was less affected by temperature than those of living (sub)tropical sea cows. This phenotype presumably safeguarded $O_2$ delivery to cool peripheral tissues and largely arises from a reduced intrinsic temperature sensitivity of the *H. gigas* protein. Additional experiments on *H. gigas* $\beta/\delta82Asn\rightarrow Lys$ mutant Hb further reveal this exchange renders Steller's sea cow Hb unresponsive to the potent intraerythrocytic allosteric effector 2,3-diphosphoglycerate, a radical modification that is the first documented example of this phenotype among mammals. Notably, $\beta/\delta82Lys\rightarrow Asn$ moreover underlies the secondary evolution of a reduced blood-$O_2$ affinity phenotype that would have promoted heightened tissue and maternal/fetal $O_2$ delivery. This conclusion is bolstered by analyses of two Steller's sea cow prenatal Hb proteins (Hb Gower I; $\zeta_2\epsilon_2$ and HbF; $\alpha_2\gamma_2$) that suggest an exclusive embryonic stage expression pattern, and reveal uncommon replacements in *H. gigas* HbF ($\gamma38Thr\rightarrow Ile$ and $\gamma101Glu\rightarrow Asp$) that increased Hb-$O_2$ affinity relative to dugong HbF. Finally, the $\beta/\delta82Lys\rightarrow Asn$ replacement of the adult/fetal protein is shown to increase protein solubility, which may have elevated red blood cell Hb content within both the adult and fetal circulations and contributed to meeting the elevated metabolic (thermoregulatory) requirements and fetal growth rates associated with this species cold adaptation.

## Editor's evaluation

In this important study, the authors analyze hemoglobin (Hb) from Steller's sea cow [extinct ~250 years ago] and compare it to (sub)tropical sea cows using ancestral sequence reconstruction and site-directed mutagenesis. They convincingly show that Steller's sea cow's Hb had decreased oxygen affinity and increased solubility, indicating adaptation to cold environments. Notably, a single amino acid change accounts for most observed biochemical differences. Additionally, the discovery of a Hb insensitive to DPG adds to the significance of the findings, making this piece an interesting and informative read to all those interested in evolution and adaptation at the molecular level; after all, Hb continues to surprise us.

**eLife digest** In 1741, shipwrecked naturalist Georg Wilhelm Steller made detailed observations of large marine mammals grazing on seaweed in the shallow waters surrounding a remote island in the North Pacific Ocean. Within thirty years, these 'Steller's sea cows' had been hunted to extinction. Unlike their remaining tropical relatives – dugongs and manatees – Steller's sea cows were specialized to cold, sub-Arctic environments. Measuring up to 10 meters long, they were much larger than other sea cow species. This, along with having very thick skin, helped them to reduce heat loss.

Previous work showed that the hemoglobin protein – which binds to and carries oxygen around mammalian bodies – of Steller's sea cows had a decreased affinity for oxygen, resulting in greater delivery of oxygen to organs and tissues. It was thought that this could be an adaptation to fuel heightened metabolic heat production in cold conditions. Studies of ancient DNA also identified the substitution of a single building block in the Steller's sea cow hemoglobin protein that is not present in other mammals and was suspected to underlie this modification.

To determine how this unique substitution affects Steller's sea cow hemoglobin function – and whether it contributed to their ability to live in cold environments – Signore et al. generated hemoglobin proteins of Steller's sea cows, dugongs and Florida manatees. Testing their biochemical properties showed that this single exchange profoundly alters multiple aspects of how the Steller's sea cow hemoglobin works.

Alongside reducing hemoglobin's oxygen affinity, the Steller's sea cow substitution also makes the protein more soluble, potentially increasing the level of hemoglobin within red blood cells. Additionally, it eliminates hemoglobin sensitivity to a molecule involved in oxygen binding – known as DPG – saving energy by no longer requiring production of this molecule. Furthermore, the same substitution makes hemoglobin less sensitive to changes in temperature, which would have helped to safeguard the delivery of oxygen to cool limbs and other extremities, reducing costly heat loss. Together, these changes in hemoglobin would have helped the Steller's sea cow to more efficiently transport oxygen around the body. Importantly, generating and testing Steller's sea cow pre-natal hemoglobins suggested this substitution may have also helped to enhance the fetal growth rate of these immense marine mammals by improving gas exchange between the mother and fetus.

Signore et al. have revealed how a mutated form of hemoglobin allowed an extinct mammal to adapt to an extreme environment. Similar methods could be used to understand the physiological attributes of other extinct animals. In the future, this increased understanding of hemoglobin mutations could aid the development of human hemoglobin substitutes for therapeutic uses.

## Introduction

The underwater foraging time of mammals is dictated by onboard oxygen stores and the efficiency of their use. Thus, evolutionary increases in oxygen stores, in the form of increased hemoglobin (Hb) and myoglobin—located within erythrocytes and skeletal/cardiac muscle, respectively—are nearly ubiquitous among mammalian divers (**Ponganis, 2011**). Notable exceptions to this rule are extant sirenians (sea cows), a group of strictly aquatic, (sub)tropical herbivores encompassing only four members; three species of manatee (family Trichechidae), and the dugong, *Dugong dugon* (family Dugongidae). While sirenians are proficient divers, they do not exhibit the greatly elevated body $O_2$ stores or an enhanced dive reflex common to other lineages of marine mammals (**Blessing, 1972**; **Scholander and Irving, 1941**). Rather, previous work revealed that the sirenian's secondary transition to aquatic life coincided with a rapid evolution of their Hb encoding genes due, in part, to gene conversion events with a neighboring globin pseudogene (**Signore et al., 2019**). The resulting high blood-$O_2$ affinity phenotype presumably allows extant sea cows to maximize $O_2$ extraction from the lungs during submergence at the cost of somewhat reduced $O_2$ offloading, thus lowering the overall metabolic intensity and fostering a prolonged breath-hold capacity (**Signore et al., 2019**).

While the relatively limited thermoregulatory capacity of extant sea cows confines them to (sub) tropical waters (**Gallivan et al., 1986**; **Marsh et al., 2011**), fossil evidence and first-hand accounts of the sub-Arctic Steller's sea cow (*Hydrodamalis gigas*) provide insights into the biological and morphological adaptations of this titanic sirenian to the harsh coastal conditions of the North Pacific, where they persisted from the Miocene (5–8 million years ago) until their demise in 1768 (**Charache et al.,**

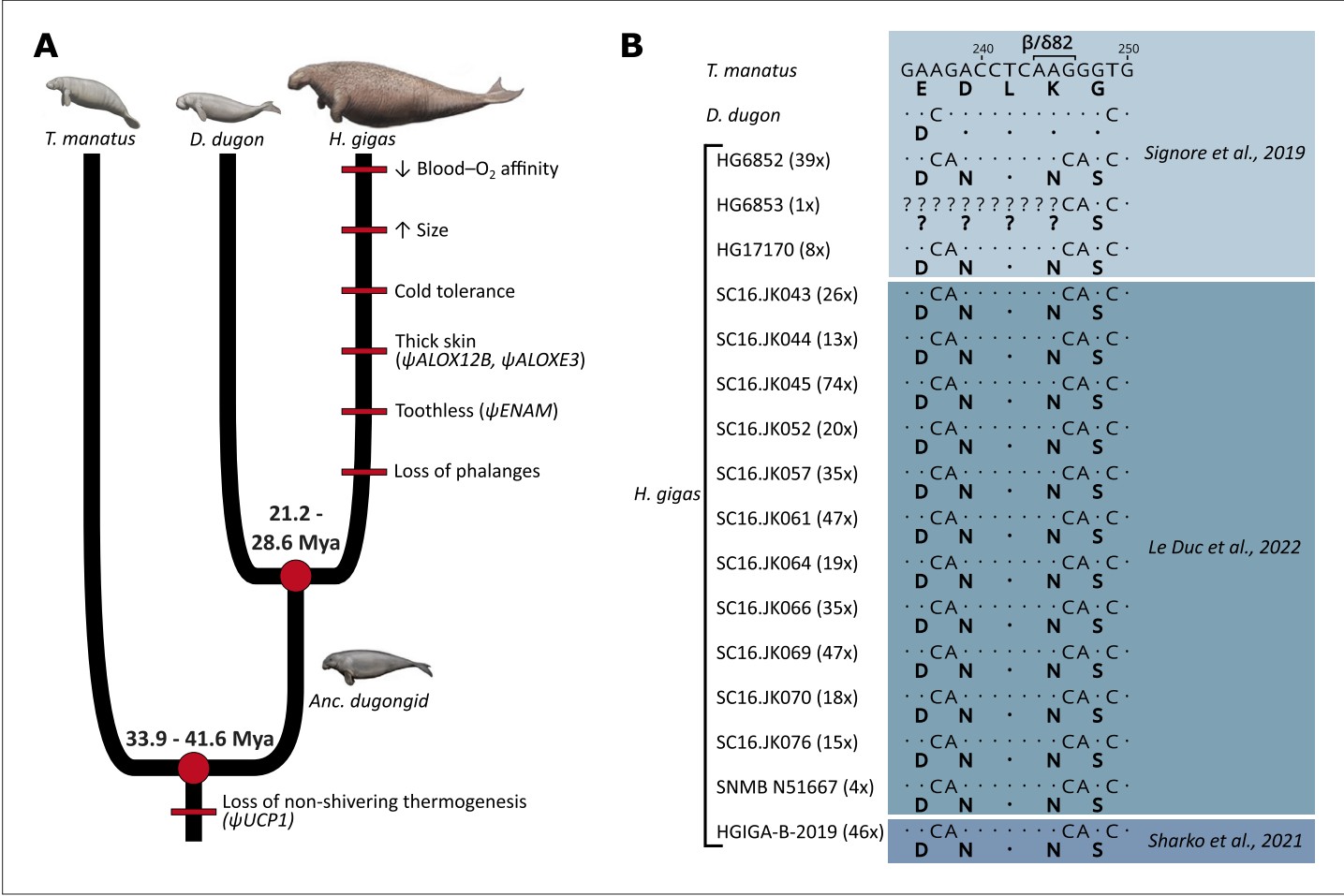

**Figure 1.** Evolution of notable morphological and genetic attributes within Sirenia. (**A**) Phenotypic innovations contributing to the unique biology of Steller's sea cows are mapped along the sirenian phylogeny (red bars) with the underlying genetic causes shown in brackets where known. Note that the red bars do not represent the dating of these traits and that their placement order is arbitrary. Divergence dates in millions of years (Mya) are based on and **Heritage and Seiffert, 2022**. The ancestral dugongid (Anc. dugongid) is represented by a late-Oligocene *Metaxytherium spp*. Sirenian paintings by Carl Buell are adapted from Figure 3 of **Springer et al., 2015** and are used with permission of J. Gatesy. (**B**) Partial nucleotide alignment of the sirenian β/δ-globin gene encoding the central region of the 2,3-diphosphoglycerate binding pocket of hemoglobin; corresponding amino acid residues (bolded) are provided below each sequence. The G→C nucleotide mutation underlying the otherwise invariant amino acid substitution (β/δ82Lys→Asn; K→N) of Steller's sea cow (*Hydrodamalis gigas*) hemoglobin is apparent in all 16 individuals for which sequence data is available. Values in brackets next to each *H. gigas* specimen represent the depth of sequence coverage for this nucleotide. Dots represent sequence identity with the Florida manatee (*Trichechus manatus*).

The online version of this article includes the following figure supplement(s) for figure 1:

**Figure supplement 1.** Amino acid sequences of sirenian *HBA* and *HBB/HBD* genes and the reconstructed sequences of the last common ancestor ('Anc. dugongid') shared by the dugong (*Dugong dugon*) and Steller's sea cow (*Hydrodamalis gigas*).

---

*1978*; **Domning, 1976**; **Heritage and Seiffert, 2022**; **Stejneger, 1887**; **Steller, 1751**). The retrieval of ancient genetic material from museum specimens has since been instrumental in clarifying the phylogenetic affinities and population history of this species, while providing additional details regarding the evolution of key morphological and physiological attributes (**Gaudry et al., 2017**; **Le Duc et al., 2022**; **Mirceta et al., 2013**; **Sharko et al., 2019**; **Sharko et al., 2021**; **Signore et al., 2019**; **Springer et al., 2015**; **Figure 1A**). For example, pilot experiments on 'resurrected' Steller's sea cow recombinant Hb demonstrated that the Hb-O$_2$ affinity of this lineage secondarily decreased following their divergence from dugongs between the mid-Oligocene and early Miocene (**Signore et al., 2019**). While sirenians do not possess the capacity for non-shivering thermogenesis due to pseudogenization of the *UCP1* gene (**Gaudry et al., 2017**), the reduced Hb-O$_2$ affinity shift in Steller's sea cow Hb was speculated to have promoted increased O$_2$ offloading to fuel increased thermogenesis to help cope with exposure

to cold sub-Arctic waters. Although the Hb of this species accumulated 11 amino acid replacements since its divergence from the dugong (*Figure 1—figure supplement 1*), this decrease in Hb-O$_2$ affinity was hypothesized to arise from a highly unusual 82Lys→Asn exchange in the chimeric beta-type (β/δ) chain (*Signore et al., 2019*). Data mined from more recent ancient DNA studies (*Le Duc et al., 2022*; *Sharko et al., 2021*) confirms that this substitution was fixed in the last remaining Steller's sea cow population (*Figure 1B*), though the specific functional effect(s) of this substitution have not been characterized. This replacement is intriguing not only because β82Lys is invariant among characterized mammalian Hbs, but because human variants with substitutions at this position display profound alterations in both structural and functional properties (*Abraham et al., 2011*; *Bonaventura et al., 1976*; *Ikkala et al., 1976*; *Sugihara et al., 1985*). For example, the human Hb Providence (β82Lys→Asn) variant exhibits a decreased inherent Hb-O$_2$ affinity and markedly reduced sensitivity to the allosteric effectors 2,3-diphosphoglycerate (DPG), Cl$^-$, and H$^+$ (*Abraham et al., 2011*; *Bardakjian et al., 1985*; *Bonaventura et al., 1976*; *Charache et al., 1977*; *Weickert et al., 1999*), all of which preferentially bind and stabilize the (low O$_2$ affinity) deoxy-state conformation of the protein. However, opposite to what was suggested for Steller's sea cows (*Signore et al., 2019*), this exchange causes the whole blood O$_2$ affinity of Hb Providence carriers to be noticeably higher than that of the general population (*Bardakjian et al., 1985Bonaventura et al., 1976*; *Charache et al., 1977*). It is thus unclear if and how the Steller's sea cow β/δ82Lys→Asn replacement underlies the lower Hb-O$_2$ affinity of this extinct species relative to other sirenians, or whether this attribute arises from one (or more) of the other 10 residue replacements that evolved in this lineage.

The β/δ82Lys→Asn exchange also raises other evolutionary significant questions, as it is predicted to have altered multiple aspects of Hb function that may lead to antagonistic pleiotropic effects. Notably, this exchange may detrimentally increase the effect of temperature on Hb-O$_2$ binding and release (*Signore et al., 2019*). The formation of the weak covalent bond between O$_2$ and the heme iron requires free energy, thus dictating an inverse relationship between Hb-O$_2$ affinity and temperature (*Weber and Campbell, 2011*). In temperate and Arctic endotherms this inherent attribute of Hb potentially impedes O$_2$ delivery to the limbs and flukes, which are maintained at substantively lower temperatures to minimize heat loss and hence energy requirements (*Campbell and Hofreiter, 2015*). Accordingly, heterothermic mammals generally possess Hbs whose O$_2$ binding properties are less sensitive to temperature than the Hbs of non-cold-adapted species, thereby maintaining sufficient O$_2$ offloading in the face of decreasing tissue temperatures. This reduction in thermal sensitivity (quantified as the overall enthalpy of oxygenation, ΔH'), appears to predominantly arise from an increased interaction between allosteric effectors and the Hb moiety (an exothermic process), which releases additional heat to assist with deoxygenation (*Weber and Campbell, 2011*). Hence, the *H. gigas* β/δ82Lys→Asn replacement, which deletes integral binding sites for the heterotropic ligands Cl$^-$ and DPG (*Bonaventura et al., 1976*), is puzzling in that it is expected to maladaptively increase the effect of temperature on O$_2$ uptake and release in the blood of the sub-Arctic Steller's sea cow.

Taken together, it remains unknown whether the *H. gigas* β/δ82Lys→Asn residue exchange contributed adaptively to the species biology or is instead linked to small population sizes (e.g. genetic drift) over the past half million years (*Le Duc et al., 2022*; *Sharko et al., 2021*). To unravel the combined effects of evolved amino acid replacements on hemoglobin function in relation to the extreme thermal biology of the extinct Steller's sea cow, we synthesized recombinant Hb proteins of this extinct species together with those of the extant dugong (*Dugong dugon*) and Florida manatee (*Trichechus manatus latirostris*), and measured their O$_2$ binding properties, relative solubility, responses to allosteric effectors, and thermal sensitivities. We also synthesized a *H. gigas* β/δ82Asn→Lys Hb mutant to assess the specific effects of this exchange, together with the Hb of the last common ancestor ('ancestral dugongid') shared between the dugong and Steller's sea cow (*Figure 1A*) in order to assess the directionality of evolved physicochemical changes in Hb function.

## Results and discussion
### O$_2$ affinity of sirenian Hbs
Measured O$_2$-equilibrium curves of the five examined Hbs revealed marked differences in intrinsic O$_2$ affinity (*Figure 2A*, *Figure 2—figure supplements 1–2*, and *Supplementary file 1a*). In the absence of allosteric effectors (pH 7.2, 37°C), the P$_{50}$ (the O$_2$ tension resulting in 50% Hb-O$_2$ saturation) of Steller's

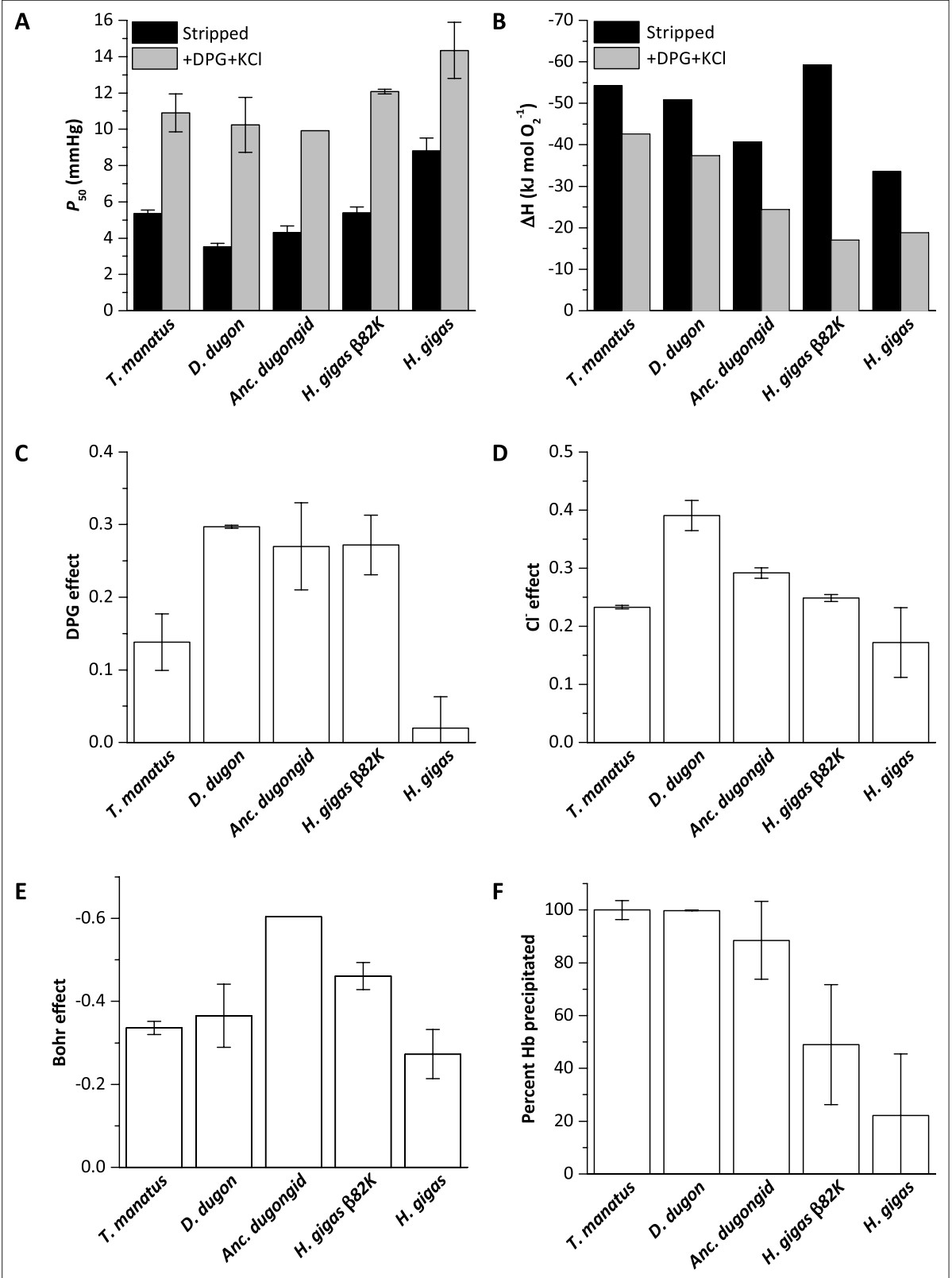

**Figure 2.** Biochemical properties of hemoglobins (Hb) from manatee (*Trichechus manatus*), dugong (*Dugong dugon*), ancestral dugongid (Anc. dugongid), Steller's sea cow (*Hydrodamalis gigas*), and a Steller's sea cow β/δ82Asn→Lys mutant (*H. gigas* β82K). All values were measured at 37°C and corrected to pH 7.2, with error bars representing the standard error of the regression estimate. (**A**) Oxygen tensions at half $O_2$ saturation ($P_{50}$) in the absence (stripped) and presence of allosteric cofactors (twofold molar excess of 2,3-diphosphoglycerate (DPG) and 0.1 M KCl). (**B**) The enthalpy of

*Figure 2 continued on next page*

*Figure 2 continued*

oxygenation (ΔH) between 25 and 37°C in stripped Hb and in the presence of allosteric cofactors (twofold molar excess DPG and 0.1 M KCl). (**C**) The effect of DPG on sirenian Hbs determined from $\log P_{50}^{(0.5 \text{ mM DPG})}$-$\log P_{50}^{(stripped)}$. (**D**) The effect of chloride on sirenian Hbs determined from $\log P_{50}^{(0.1 \text{ M KCl})}$-$\log P_{50}^{(stripped)}$. (**E**) The Bohr effect of sirenian Hbs in the presence of allosteric cofactors (twofold molar excess 2,3-diphosphoglycerate (DPG) and 0.1 M KCl), as calculated from $\Delta \log P_{50}/\Delta pH$ over the pH range 6.9 and 7.8. (**F**). The relative solubility of sirenian Hbs is denoted by the percentage of Hb protein precipitated by the addition of 3 M ammonium sulfate.

The online version of this article includes the following source data and figure supplement(s) for figure 2:

**Source data 1.** Source data for *Figure 2*.

**Figure supplement 1.** The pH dependence of oxygen tensions and the cooperativity coefficients at half $O_2$ saturation ($P_{50}$ and $n_{50}$, respectively) for hemoglobins of the Florida manatee (*Trichechus manatus latirostris*), dugong (*Dugong dugon*), Steller's sea cow (*Hydrodamalis gigas*), and the last common dugonid ancestor ('Anc. dugongid') in stripped Hb (circles), and in the presence of 0.1 M KCl (triangles), of a twofold molar excess of 2,3-diphosphoglycerate (DPG; inverted triangles), and of both KCl and DPG (squares), at 25°C (solid symbols) and 37°C (open symbols).

**Figure supplement 1—source data 1.** Source data for *Figure 2—figure supplement 1*.

**Figure supplement 2.** The pH dependence of oxygen tensions and the cooperativity coefficients at half $O_2$ saturation ($P_{50}$ and $n_{50}$, respectively) of wild-type Steller's sea cow hemoglobin (*H.gigas W-T*) and a mutated Steller's sea cow β/δ82Lys variant (*H.gigas β82*K) in the absence and presence of allosteric effectors (0.1 M KCl and/or twofold molar excess of 2,3-diphosphoglycerate (DPG)) at 25°C (solid symbols) and 37°C (open symbols).

**Figure supplement 2—source data 1.** Source data for *Figure 2—figure supplement 2*.

**Figure supplement 3.** Solubility assay of five sirenian hemoglobins (Hb) illustrating the percentage of Hb protein remaining in solution after precipitation by the addition of ammonium sulfate.

**Figure supplement 3—source data 1.** Source data for *Figure 2—figure supplement 3*.

sea cow Hb ($P_{50}$=8.8 mm Hg) is ~2 times higher than that of dugong (3.5 mm Hg), ancestral dugongid (4.3 mm Hg), and manatee (5.4 mm Hg) Hbs under the same conditions (*Figure 2A*, *Figure 2—figure supplement 1*, and *Supplementary file 1a*). Site-directed mutagenesis experiments reveal that the increased intrinsic $P_{50}$ of Steller's sea cow Hb predominantly arises from the β/δ82Asn substitution, as the β/δ82Asn→Lys mutant exhibits an intrinsic $P_{50}$ similar to that of the ancestral dugongid (*Figure 2A*, *Figure 2—figure supplement 2*). Of note, the $O_2$ affinity of dugong, ancestral dugongid, and manatee Hbs was reduced in the presence of $Cl^-$ and DPG ($P_{50}$=10.2, 9.9, and 10.9 mm Hg, respectively) by a similar degree to that of Asian elephant Hb (*Campbell et al., 2010a*). This finding extends previous studies conducted on sirenian Hbs (*Farmer et al., 1979*; *McCabe et al., 1978*; *Signore et al., 2019*), and reveals that the high $O_2$ affinity of dugong and manatee blood is not attributable to decreased allosteric effector sensitivity. Conversely, Steller's sea cow Hb was shown to be markedly less responsive to allosteric effectors, as only a moderate reduction in $O_2$-affinity was observed in the presence of $Cl^-$ and DPG ($P_{50}$=14.3 mm Hg). When the effects of these allosteric effectors were measured individually, Steller's sea cow Hb exhibits lower DPG, $Cl^-$, and $H^+$ (Bohr) effects relative to those of the ancestral dugongid and β/δ82Asn→Lys mutant (*Figure 2C–E*, *Supplementary file 1a*). These data confirm that a high intrinsic (i.e. in the absence of allosteric effectors) Hb-$O_2$ affinity is an ancient sirenian trait that likely aided the transition of the group to the aquatic environment, and that Hb-$O_2$ (and hence whole blood) affinity was secondarily reduced in the Steller's sea cow lineage (*Signore et al., 2019*). This latter finding contrasts with allometric expectations for mammals—whereby blood $O_2$ affinity and body mass are inversely correlated (*Schmidt-Neilsen and Larimer, 1958*)—and thus further suggests this modification served an adaptive function in this extinct species.

The single most distinct feature of *H. gigas* Hb is the lack of a discernable effect of DPG on $P_{50}$ ($\Delta \log P_{50}^{(DPG-stripped)}$=0.02 at 37 °C and pH 7.2; *Figure 2C*, *Supplementary file 1a*), relative to the Hbs of the ancestral dugongid (0.27) and the extant manatee and dugong (0.14 and 0.30, respectively). This intracellular effector generally occurs in equimolar concentrations to Hb (*Bunn, 1980*) and strongly decreases the $O_2$ affinity of most mammalian Hbs via direct electrostatic interactions with β2His and β82Lys (*Figure 3A*), together with potential water-mediated interactions with β143His and the α-$NH_2$ group of 1Val of the $\beta_2$ chain (*Richard et al., 1993*). However, unlike the other (ionizable) residues whose ability to interact with DPG is highly pH dependent, β82Lys is strongly cationic and thus is able to bind DPG across the entire physiological pH range. Presumably arising from this indispensable role in DPG binding, this residue is uniformly conserved in mammalian Hbs (*Figure 3A*), with the exception of several heterozygous adult human HbA carriers with substitutions at this position (*Ikkala et al., 1976*; *Moo-Penn et al., 1976*; *Sugihara et al., 1985*). Given that none of the other six

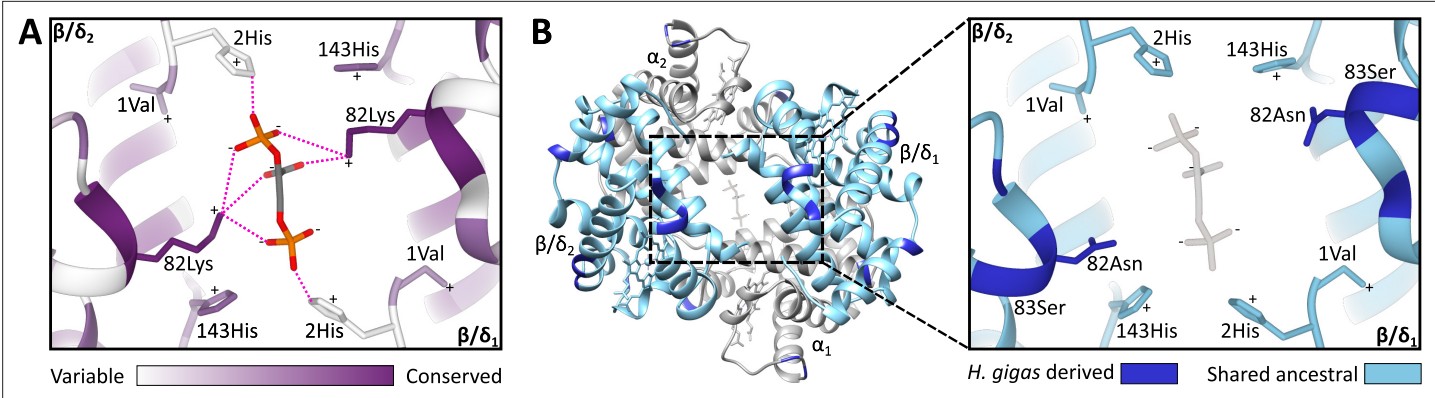

**Figure 3.** Homology models of the 2,3-diphosphoglycerate (DPG) binding site in ancestral dugongid and Steller's sea cow (*Hydrodamalis gigas*) hemoglobin. (**A**) Model of the ancestral dugongid DPG binding site. Amino acids are colored according to the degree of sequence conservation. Notably, β/δ82Lys shows the highest level of sequence conservation, as it is able to bind to multiple sites on the DPG molecule (indicated by dashed pink lines), whereas β/δ2His is only able to directly interact with DPG in the protonated state. (**B**) Model of Steller's sea cow hemoglobin (left) and a close-up of the DPG binding pocket (right). Dark blue colored residues represent the 11 *H. gigas* specific substitutions, while those in light blue denote the ancestral state. Homology modeling illustrates how the replacement of β/δ82Lys with neutral Asn inhibits DPG binding to the hemoglobin molecule.

The online version of this article includes the following figure supplement(s) for figure 3:

**Figure supplement 1.** Homology model of Steller's sea cow (*Hydrodamalis gigas*) adult-expressed ($\alpha_2\beta/\delta_2$) hemoglobin with (A) the β/δ-subunits (blue) in the foreground and (B) the α-subunits (gray) in the foreground.

β/δ-chain replacements that evolved on the Steller's sea cow branch (*Figure 3B*, *Figure 3—figure supplement 1*, *Figure 1—figure supplement 1*) are implicated in DPG binding, the deletion of the integral DPG binding site at β/δ82 in Steller's sea cow Hb is fully consistent with its inability to bind DPG (*Figure 3B*). This conclusion is further supported by our measurements on the Steller's sea cow β/δ82Asn→Lys mutant, which shows that reversion to the ancestral state restores the DPG effect to the same level observed in ancestral dugongid Hb (*Figure 2C*). Notably, and despite possessing the identical DPG binding site residues as Hb Providence, the *H. gigas* protein exhibits a distinctly lower DPG effect than this human variant ($\Delta logP_{50}^{(DPG-stripped)}$=0.08; *Bonaventura et al., 1976*; *Bardakjian et al., 1985*). The lower DPG sensitivity of Steller's sea cow Hb thus implicates an epistatic contribution from other amino acids in the vicinity of the DPG pocket. Importantly, the Lys→Asn replacement in the DPG binding pocket causes the $O_2$ affinity of human HbA to increase in the presence of allosteric cofactors (*Bardakjian et al., 1985*; *Charache et al., 1977*), whereas results presented in *Figure 2A* show Steller's sea cow Hb-$O_2$ affinity is reduced relative to its ancestors carrying β/δ82Lys under all test conditions. This observation highlights a growing body of research indicating that both the direction and overall phenotypic effect of specific amino acid substitutions may be conditional on the genetic background in which they occur (*Natarajan et al., 2023*; *Storz, 2016*).

The insensitivity of *H. gigas* Hb to DPG is also notable as it would have markedly reduced their capacity to modulate blood-$O_2$ affinity in vivo (e.g. seasonally), and is the first demonstrated example of a genuine DPG insensitive Hb phenotype among mammals. While eastern moles (*Scalopus aquaticus*) are a possible exception (*Campbell et al., 2010b*), feliform carnivores, ruminants, and two species of lemurs have also traditionally been placed in the 'DPG insensitive' category (*Bunn, 1980*) despite the fact their Hbs are moderately responsive to DPG in the absence of $Cl^-$ (*Bonaventura et al., 1976*; *Fronticelli et al., 1988*; *Janecka et al., 2015*; *Perutz et al., 1993*). Nonetheless, red blood cell DPG concentrations of species with suppressed DPG sensitivities are markedly reduced relative to mammals whose Hb-$O_2$ affinity is regulated by DPG (<0.1–1.0 mM vs. 4–10 mM, respectively; *Bunn, 1980*). While the potential benefits of a low DPG sensitivity phenotype have been debated (*Campbell et al., 2010b*; *Kay, 1977*), approximately 20% of glucose uptake by human erythrocytes is diverted to DPG synthesis via the Rapport-Luebering shunt (*Duhm et al., 1968*), thereby bypassing production of both ATP molecules generated via the anaerobic substrate level phosphorylation pathway (*Bunn, 1980*; *Kauffman et al., 2002*; *Rapoport and Luebering, 1950*). Accordingly, since each molecule of DPG produced comes at the expense of producing an ATP molecule, the probability that Steller's sea cow blood similarly contained low levels of this organophosphate is high.

## Hb solubility

Ectopic expression of human Hb Providence β82Asp mutants in *E. coli* has been shown to increase soluble protein production by 47–116% relative to the expression of human Hb variants not carrying this substitution (*Weickert et al., 1999*). Consistent with this observation, we found that Steller's sea cow Hb is more soluble than those of other sirenians and the engineered *H. gigas* β/δ82Asn→Lys mutant (*Figure 2F* and *Figure 2—figure supplement 3*). While the precise mechanism underlying this phenomenon is unknown, Hb Providence variants exhibit sharp reductions in irreversible oxidative damage of nearby β93Cys that initiates Hb denaturation (*Abraham et al., 2011*; *Jana et al., 2020*; *Strader et al., 2017*). These β82 replacements thereby presumably decrease the rate of Hb turnover and increase the half-life of the protein (*Strader et al., 2017*), which may contribute to the mild polycythemia in humans carrying this substitution (*Bardakjian et al., 1985*; *Moo-Penn et al., 1976*). Blood with an elevated $O_2$ carrying capacity is typical of most mammalian divers, where it increases onboard $O_2$ stores and extends dive times (*Ponganis, 2011*), but is not observed in extant sirenians (*Farmer et al., 1979*; *White et al., 1976*; *Wong et al., 2018*). However, any solubility-driven increases in red blood cell Hb concentration resulting from the β82Lys→Asn exchange would have allowed Steller's sea cows to maintain an elevated rate of tissue $O_2$ delivery to meet their metabolic demands during extended underwater foraging intervals. Although this species was presumably unable to completely submerge (*Domning, 2022*; *Steller, 1751*), this conjecture is corroborated by Steller's account that, *'they keep their heads always under water* [foraging]*, without regard to life and safety'* (*Steller, 1751*). A reduced potential for Hb oxidation may also help explain Steller's vexing observation that 'what is remarkable, even in the hottest days it [the flesh] can be kept in the open air for a very long time without any bad odor, even though all full of worms [maggots]' (*Steller, 1751*).

## Thermal sensitivity

The invariant energy change associated with forming the weak covalent bond between $O_2$ and the heme iron (i.e. the enthalpy of heme oxygenation; $\Delta H^{O2}$) is exothermic (–59 kJ mol$^{-1}$ $O_2$) (*Atha and Ackers, 1974*), and only moderately opposed by the endothermic solubilization of $O_2$ ($\Delta H^{H2O}$; 12.55 kJ mol$^{-1}$ $O_2$), resulting in an inverse relationship between temperature and Hb-$O_2$ affinity. However, the heat of the T→R conformational change ($\Delta H^{T \rightarrow R}$), and the oxygenation-linked dissociation of H$^+$ ($\Delta H^{H+}$), Cl$^-$ ($\Delta H^{Cl-}$), and DPG ($\Delta H^{DPG}$) may offset this relationship, such that the overall enthalpy of Hb oxygenation ($\Delta H'$) can become greatly minimized or even endothermic (*Weber and Campbell, 2011*; *Weber et al., 2010*). By facilitating adequate oxygenation of cool peripheral tissues, Hbs with numerically low $\Delta H'$ values is interpreted to be adaptive for cold-tolerant, regionally heterothermic mammals. The evolution of this phenotype has predominantly been attributed to the formation of additional heterotropic ligand binding sites on the protein moiety, as has previously been demonstrated for the woolly mammoth, *Mammuthus primigenius* (*Campbell et al., 2010a*; *Weber and Campbell, 2011*). Conversely, Steller's sea cow Hb lacks heterotropic binding of DPG and displays lower Bohr (H$^+$) and Cl$^-$ effects than all other sirenian Hbs measured (*Figure 2* and *Supplementary file 1a*), yet exhibits a $\Delta H'$ value (–18.8 kJ mol$^{-1}$ $O_2$; *Figure 2B*) that is close to that of mammoth Hb (–17.2 kJ mol$^{-1}$ $O_2$) (*Campbell et al., 2010a*). This striking convergence largely arises from the inherently low $\Delta H$ of stripped Steller's sea cow Hb (–34.2 kJ mol$^{-1}$ $O_2$) relative to dugong, ancestral dugongid, and manatee Hbs (range: –50.1 to –58.6 kJ mol$^{-1}$ $O_2$) at pH 7.8—where oxygenation-linked binding of protons is minimal—and indicates that structural differences modifying the T→R transition largely underlie the low thermal sensitivity of *H. gigas* Hb. Recent studies have shown that a large positive $\Delta H^{T \rightarrow R}$ may similarly contribute to the low $\Delta H'$ of deer mouse, cow, shrew, and mole Hbs (*Campbell et al., 2010b*; *Campbell et al., 2012*; *Jensen et al., 2016*; *Signore et al., 2012*; *Weber et al., 2014*) suggesting that this potential mechanism of temperature adaptation may be more widespread than previously appreciated. Our experiments with the Steller's sea cow β82Asn→Lys mutant further implicate substitutions at this position as a key factor underlying the inherently low $\Delta H$ of the protein, as this modified protein displays a greatly increased $\Delta H$ in the absence of allosteric effectors relative to the wild-type *Hydrodamalis* protein (*Figure 2B*). Interestingly, despite these inherent $\Delta H$ differences between the mutant and wild-type Steller's sea cow Hbs, their $\Delta H'$ values are indistinguishable in the presence of allosteric effectors (*Figure 2B*). These data suggest that β82Asn uncouples thermal sensitivity from DPG concentration, permanently conferring the *H. gigas* protein with a numerically low $\Delta H'$

by genetic assimilation while simultaneously eliminating the energetic cost of DPG production within the red blood cells.

Given both the marked functional changes observed for *H. gigas* Hb and the correspondingly large ecological and thermal shifts encountered by ancestral hydrodamaline sea cows following their exploitation of the North Pacific in the Miocene (*Heritage and Seiffert, 2022*), it is surprising that previous work failed to provide evidence for positive selection or an accelerated amino acid substitution rate for any globin loci in the Steller's sea cow branch (*Signore et al., 2019*). However, this result is not unique to hydrodamalines, as the Hb coding genes of woolly mammoths and stem penguins also lack statistically significant signatures of positive selection accompanying their niche transitions despite clear directional changes in their Hb properties (*Campbell et al., 2010a*; *Signore et al., 2021*).

## Paleophysiology of Steller's sea cows

The posthumously published behavioral and anatomical accounts of the last remaining *H. gigas* population by naturalist Georg Wilhelm Steller while stranded on Bering Island (55°N, 166°E) in 1741/1742 provide a rich tapestry to interpret the paleophysiology of this colossal marine herbivore. For example, their protective thick bark-like hide and extensive blubber layer give credence to the extreme nature of their shallow rocky and (during winter) ice-strewn habitat (*Le Duc et al., 2022*). Here, as Steller (*Steller, 1751*) remarked, they used fingerless, bristle-covered forelimbs for support and to shear *'algae and seagrasses from the rocks,'* which they masticated *'not with teeth, which they lack altogether, but with'* large, ridged keratinized pads located on the upper palate and lower mandible. Although they became visibly thin during winter when *'their spinous processes can be seen,'* Steller (*Steller, 1751*) noted that '(*t*)*hese animals are very voracious, and eat incessantly'* such that their stupendous stomach (*'6 feet* [1.8 m] *long, 5 feet* [1.5 m] *wide'*) and enormous intestines—which measured a remarkable 5958 inches (~151.5 m) from the esophagus to anus, equivalent to *'20½ times as long as the whole animal'*—is constantly *'stuffed with food and seaweed.'* The proportionally larger gut (*Domning, 2022*) is consistent with Steller's sea cow's higher energetic requirements relative to extant manatees, which, owing to their low metabolic intensities become cold-stressed and die if chronically exposed to water temperatures below 15 °C (*O'Shea et al., 1985*). The inferred reduction in insulative blubber thickness of *H. gigas* during the winter months would likely have compounded the rate of heat loss to sub-zero degree Celsius air and water, though may have been compensated for by arteriovenous anastomoses that regulated blood flow to the skin, and by countercurrent rete supplying the flippers and tail flukes, the latter of which are well developed in manatees and presumably other sirenians (*Marshall et al., 2022*; *Rommel and Caplan, 2003*). These structures conserve thermal energy by promoting profound cooling at the appendages and periphery (*McCabe et al., 1978*), and presumably underlie the low thermal dependence of Steller's sea cow Hb relative to those of extant sea cows.

Reductions in blood-$O_2$ affinity accompanying the *H. gigas* β/δ82Lys→Asn substitution are expected to have further augmented tissue $O_2$ delivery without tangible effects on lung $O_2$ uptake, thereby helping to fuel thermogenesis and maintain a stable core temperature. In the absence of UCP1-dependent nonshivering thermogenesis (*Gaudry et al., 2017*), the latter was presumably supplemented by a substantive heat increment arising from fermentation and other post-prandial processes (*Marshall et al., 2022*). Although the attendant increase in the rate of $O_2$ consumption would have mandated a reduction in breath-hold endurance—likely reflecting the relatively short submergence times (4–5 min) observed by *Steller, 1751*—our results suggest that this may have been partially counteracted by an elevated blood-$O_2$ carrying capacity that was potentially coupled to a greater lung volume (*Domning, 2022*). Underwater foraging times were presumably further defended by key components of the dive reflex, namely bradycardia and peripheral vasoconstriction. Indeed, Steller inadvertently was the first to (indirectly) describe this phenomenon as he observed his crew hunting the animals with spears and knives, *'the blood from the wounded back spurted up like a fountain. As long as he kept his head underwater the blood did not flow out, but as soon as he raised his head to breathe the blood leaped forth anew'* (*Steller, 1751*).

A final compelling aspect of Steller's sea cow evolution was their immense size—up to 11,000 kg in mass and 10 m in length—relative to extant sirenians (*Domning, 1976*). While Steller does not provide measurements of *'their tender little offspring,'* Gerhard Friedrich Müller, who edited Steller's

manuscript prior to publication, noted calves 'weighed 1200 pounds [544 kg] and upwards' (**Mueller, 1761**). This value is ~10–50 times the mass of newborn manatees and dugongs (~10–50 kg) (**Odell, 2009**) and is suggestive of rapid prenatal growth during the ~1 year gestational period indicated by Steller (**Steller, 1751**). While placental morphology and relative blood flow are important factors affecting pre-natal growth rates, the efficiency of maternal/pre-natal gas exchange is also influenced by differences in blood $O_2$-affinity between the two circulations (**Carter, 2015**). During the early stages of mammalian development, $O_2$ diffusion is optimized via the expression of embryonic Hb isoforms with high $O_2$-affinity (**Weber et al., 1987**). Briefly, the α and β gene families of mammals possess multiple paralogs, with the 5'–3' linkage order and their distance from the respective upstream locus control regions dictating the expression pattern of each locus throughout development (**Peterson and Stamatoyannopoulos, 1993**). Thus, at two weeks post-conception, developing human embryos begin expressing genes at the 5' end of the α (HBZ) and β (HBE) clusters, which are translated into ζ - and ε-globin chains, respectively, to form Hb Gower I, $\zeta_2\varepsilon_2$ (**Fantoni et al., 1981**). At week four, expression of the downstream HBA and HBG loci add α- and γ-chains to the erythrocytes of the developing circulatory system to generate additional Hb isoforms including HbF ($\alpha_2\gamma_2$) (**Hecht et al., 1966**). Notably, this pattern of gene expression switching during development results in the temporal production of Hb isoforms with successively lower $O_2$ affinities (i.e. each Hb isoform has a lower $O_2$ affinity than the protein it replaced), which facilitates $O_2$ transfer from maternal to embryonic and fetal blood (**Carter, 2015**).

In all mammalian lineages examined to date, with the exception of bovid artiodactyls (e.g. goats, sheep, and cows) and simian primates, the expression of the above Hb isoforms is thought to be limited to the embryonic stage of development (**Carter, 2015**); as such, most mammals express the same Hb isoform (HbA) during both the fetal and post-natal stages of development. However, observations suggest that sea cows and proboscideans (elephants) may also express distinct fetal isoHbs. For example, blood from a 5 month-old elephant fetus was shown to contain two distinct Hb components, although only a single (adult) Hb component is present in 12-month-old fetal and adult blood (**Riegel et al., 1967**). Likewise, the blood of manatee calves contains a second isoHb (comprising ~5% of total Hb; **Farmer et al., 1979**), which moreover appears to exhibit $O_2$ binding properties distinct from that of maternal blood (**White et al., 1976**). It is thus conceivable that the second Hb component in newborn manatee blood (and potentially other sirenians) arises from the delayed expression of HBG (which expresses the γ-chain of HbF). Given that the timing of globin gene expression is determined by its distance from the locus control region (**Peterson and Stamatoyannopoulos, 1993**), the possible attenuation of HBG expression in sirenians HBG relative to elephants is supported by synteny comparisons of the β-globin gene cluster, as the HBG locus of sirenians is further downstream than the same locus is in the elephant cluster (see Figure 1B of **Signore et al., 2019**). If expression of the sirenian HBG locus is developmentally delayed to form a discrete isoHb in fetal blood, the resulting HbF protein would be expected to display $P_{50}$ and cooperativity ($n_{50}$) values that fall between Gower I and HbA, and a response to pH that is similar to the latter.

To test this hypothesis and better understand the maternal/pre-natal gas exchange strategy of sirenians, we thus expressed recombinant Hbs corresponding to Steller's sea cow Gower I and HbF and dugong HbF (whose γ-chain differs from *H. gigas* γ at four positions **Signore et al., 2019**; **Figure 4**, **Figure 4—figure supplement 1**), and measured their $O_2$ binding properties and response to allosteric effectors. As expected, the $P_{50}$ of *H. gigas* Gower I in the combined presence of Cl⁻ and DPG is markedly less than adult Steller's sea cow Hb (3.1 vs. 15.2 mm Hg, respectively) (**Figure 4A**). Similarly, the Hb-$O_2$ affinity of Steller's sea cow and dugong HbF ($P_{50}$ of 0.56 and 1.2 mm Hg, respectively) are substantially higher than that of their respective adult counterparts and, unexpectedly, also higher than that of Steller's sea cow Gower I (**Figure 4A**). In line with the embryonic expressed Hb isoforms of other mammals (**Brittain, 2002**; **Weber et al., 1987**), the Bohr and cooperativity coefficients of the sirenian Gower I and HbF proteins were also substantially lower than that of the post-natal (HbA) isoform (**Figure 4A** and **Supplementary file 1b**). Accordingly, their functional properties are consistent with the embryonic (but not fetal) Hbs of other mammalian species. Although it remains possible that expression of these isoforms lingers into late fetal development, the upstream HBG transcriptional control motif ('CACCC') crucial for the suppression of human HBB gene expression during the fetal stage (**Perez-Stable and Costantini, 1990**) is mutated in both dugongs and Steller's sea cows (but not manatees or elephants; **Figure 4—figure supplement 2**). Consequently, the primary (if not

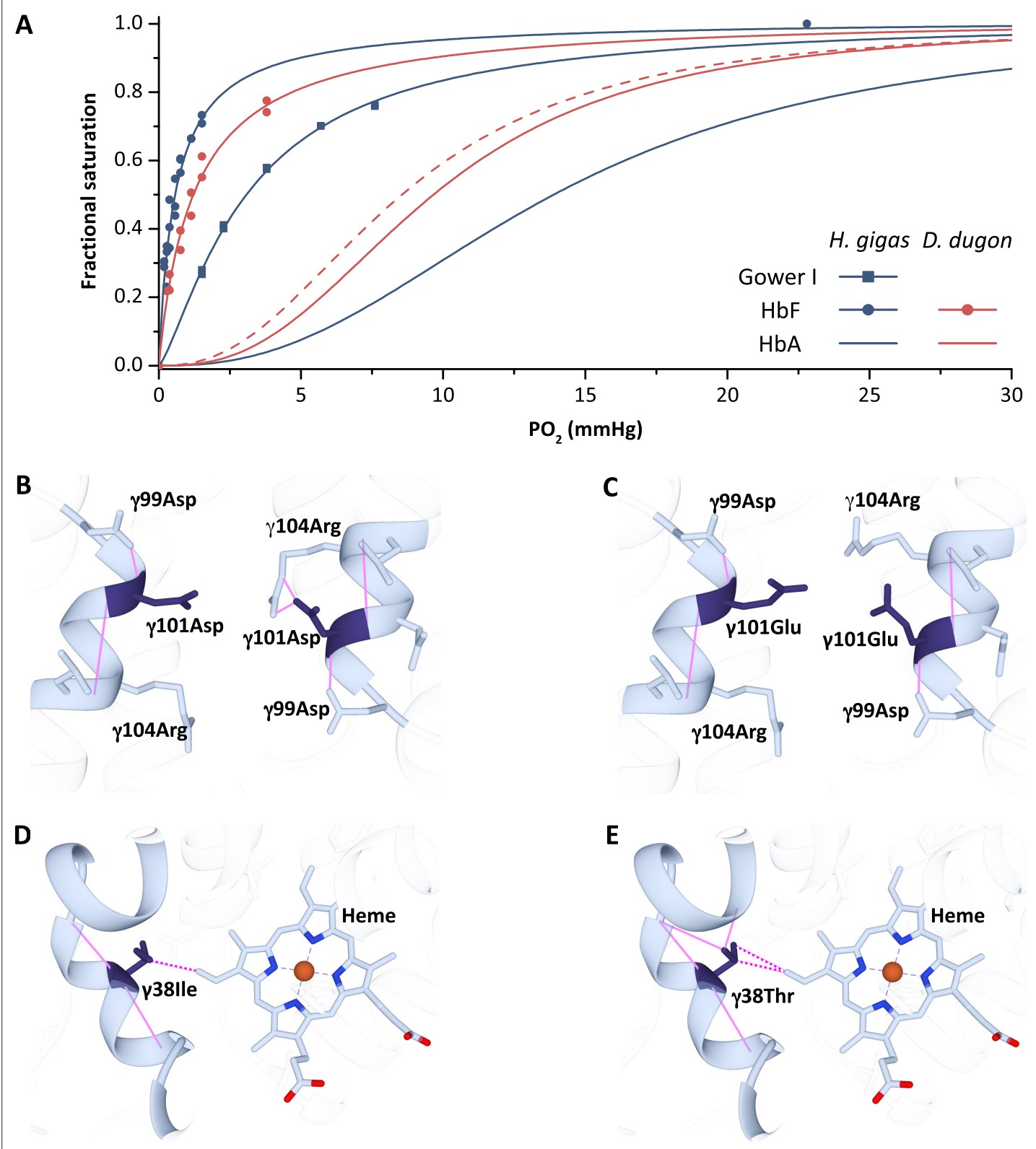

**Figure 4.** Oxygen equilibrium curves of prenatal and adult sirenian hemoglobins. (**A**) Oxygen equilibrium curves for Steller's sea cow (*Hydrodamalis gigas*; blue) hemoglobin (Hb) Gower I ($\zeta_2\varepsilon_2$), HbF ($\alpha_2\gamma_2$), and HbA ($\alpha_2\beta/\delta_2$) and dugong (*Dugong dugon*; red) HbF and HbA in the presence of allosteric cofactors 2,3-diphosphoglycerate (DPG) (twofold molar excess) and KCl (100 mM) at pH 7.1 (prenatal Hbs) or 7.2 (HbA).The dashed red line is for dugong HbA in the absence of DPG which illustrates relative differences in O$_2$ affinity between the maternal (solid line) and fetal (dashed line)

*Figure 4 continued on next page*

*Figure 4 continued*

circulations. Homology models of Steller's sea cow and dugong HbF denote structural alterations arising from the *H. gigas* specific 101 (**B**) vs. (**C**), respectively, and 38 (**D**) vs. (**E**), respectively, replacements. Solid pink lines denote predicted hydrogen bonds while the dashed pink lines represent predicted Van der Waals interactions.

The online version of this article includes the following figure supplement(s) for figure 4:

**Figure supplement 1.** Homology model of Steller's sea cow (*Hydrodamalis gigas*) HbF hemoglobin ($\alpha_2\gamma_2$) with (A) the α-subunits in the foreground and (B) the γ-subunits in the foreground.

**Figure supplement 2.** DNA alignment of the 5' untranslated region and transcriptional promoter regions for representative paenungulate and human HBG genes. Note that hyrax HBG is pseudogenized (*Signore et al., 2019*) and hence is not included in the alignment.

sole) Hb isoform expressed within both the fetal and post-natal circulation of dugongids is almost certainly HbA. Intriguingly, however, Steller's sea cow HbF exhibits a distinctly higher $O_2$ affinity but lower cooperativity than dugong HbF, traits that are likely attributed to two exceedingly rare γ-chain amino acid replacements positioned within the interior of the protein (γ38Thr→Ile and γ101Glu→Asp; *Figure 4B–E*, *Figure 4—figure supplement 1*). Briefly, the central cavity γ101Glu→Asp replacement alters the highly conserved sliding interface between the $\alpha_1\gamma_2$ dimer subunits by forming a hydrogen bond with γ104Arg (*Figure 4B*) and has been shown to increase the intrinsic affinity of human Hb Potomac (β101Glu→Asp) (*Charache et al., 1978*; *Shih et al., 1985*). Residue γ38 is also potentially functionally relevant as it is located along the $\alpha_2\gamma_1$ sliding interface and is in contact the distal heme (*Figure 4D*; *Ropero et al., 2006*). Of note, however, the mRNA capping site of *H. gigas* HbF exhibits an A→G transversion mutation (*Figure 4—figure supplement 2*) that has been shown to lower transcript levels of human HBB by twofold (*Myers et al., 1986*). It thus remains unknown if *H. gigas* HbF exhibited a similar downregulation and hence to what degree these γ-chain replacements may have altered $O_2$ transfer to the Steller's sea cow embryo through the amniotic fluid prior to placental development.

Based on the above considerations, the apparently unique DPG insensitive phenotype of *H. gigas* HbA is particularly noteworthy owing to its potential impact on maternal/fetal $O_2$ exchange. Presumably, to assist in this process, fetal blood cells expressing HbA contain only trace amounts of DPG hence conferring fetal blood with an elevated $O_2$ affinity relative to that of the maternal circulation (compare the dashed vs. solid red lines in *Figure 4A* as an example) in the vast majority of mammalian species (*Bunn, 1980*; *Carter, 2015*). By contrast, and owing to the inability of Steller's sea cows Hb to respond to DPG in either the fetal or adult circulations, this species would represent a rare example (feloids and eastern moles are others) in which fetal and maternal blood has the same (albeit relatively low) $O_2$ affinity. However, placental $O_2$ delivery in these species will be defended by the well-known double Bohr effect, whereby $CO_2$ transport from the fetal to maternal circulation increases blood $O_2$ affinity in the former while lowering it in the latter (*Carter, 2015*). More importantly, the evolved reduction in blood $O_2$ affinity of Steller's sea cows would have stipulated that equilibrium between umbilical and uterine blood was reached at a higher $PO_2$, a condition that is expected to substantially improve $O_2$ delivery to the fetal circulation (*Carter, 2015*). The lower fetal blood $O_2$ affinity (relative to manatees and dugongs) and concomitant higher fetal blood-to-tissue $PO_2$ gradients are further expected to have augmented $O_2$ delivery to the developing tissues of this species. As such, these attributes, together with increases in Hb solubility/reduced susceptibility to oxidative damage arising from β/δ82Lys→Asn that conceivably also elevated the $O_2$ carrying capacity of fetal blood, may have been important contributors to the enhanced fetal growth rate of these immense sirenians. The resulting increase in thermal inertia and relatively low surface-area-to-volume ratio following birth, together with an adaptively reduced Hb thermal sensitivity and thick 'bark-like' skin arising from the inactivation of lipoxygenase genes (*Le Duc et al., 2022*), were presumably central components of Steller's sea cows successful exploitation of the harsh sub-Arctic marine environments of the North Pacific.

## Materials and methods

### Sequence collection and analyses

The pre-natal (HBZ-T1, HBE, and HBG) and adult-expressed Hb genes (HBA and HBB/HBD) of the Florida manatee, dugong, and Steller's sea cow, and the most recent common ancestor shared by Steller's sea cow and the dugong ('ancestral dugongid') have previously been determined (*Signore et al., 2019*). As the *H. gigas* β/δ82Lys→Asn exchange is not known to occur in any living species, we mined recently deposited genomes for 13 additional Steller's sea cows (PRJNA484555, PRJEB43951) to test for the prevalence of this replacement in the population. Briefly, we first searched the SRA files of each specimen using the megablast function against a previously determined *H. gigas HBB/HBD* gene sequence (GenBank accession #: MK562081). All hits were then downloaded, trimmed of adapters and low-quality regions using BBDuk (Joint Genome Institute), and assembled to *H. gigas HBB/HBD* using Geneious Prime 2019 software (Biomatters Ltd, Auckland, New Zealand). Assemblies generated using genome reads that were not pre-treated with uracil-DNA glycosylase and endonuclease VIII to reduce C→T and G→A damage artifacts (*Le Duc et al., 2022*) were examined to ensure these deamination artifacts did not affect the consensus sequences.

### Construction of recombinant Hb expression vectors

Coding sequences for Steller's sea cow Gower I ($\zeta_2\varepsilon_2$), dugong, and *H. gigas* HbF ($\alpha_2\gamma_2$), and the above four HbA ($\alpha_2\beta/\delta_2$) proteins were optimized for expression in *E. coli* and synthesized in vitro by GenScript (Piscataway, NJ). The resulting gene cassettes were digested with restriction enzymes and tandemly ligated into a custom Hb expression vector (*Natarajan et al., 2011*) using a New England BioLabs Quick Ligation Kit as recommended by the manufacturer. Chemically competent JM109 (DE3) *E. coli* (Promega) were prepared using a Z-Competent *E. coli* Transformation Kit and Buffer Set (Zymo Research). We also prepared a *H. gigas* β/δ82Asn→Lys Hb mutant via site-directed mutagenesis on the Steller's sea cow Hb expression vector by whole plasmid amplification using mutagenic primers and Phusion High-Fidelity DNA Polymerase (New England BioLabs), phosphorylation with T4 Polynucleotide Kinase (New England BioLabs), and circularization with an NEB Quick Ligation Kit (New England BioLabs). All site-directed mutagenesis steps were performed using the manufacturer's recommended protocol.

Hb expression vectors were co-transformed into JM109 (DE3) chemically competent *E. coli* alongside a plasmid expressing methionine aminopeptidase (*Natarajan et al., 2011*), plated on LB agar containing ampicillin (100 µg/ml) and kanamycin (50 µg/ml), and incubated for 16 hr at 37°C. A single colony from each transformation was cultured in 50 ml of 2xYT broth for 16 hr at 37°C while shaking at 200 rpm. Post incubation, 5 ml of the culture was pelleted by centrifugation, and plasmid DNA was isolated using a GeneJET Plasmid Miniprep Kit (Thermo Scientific). The plasmid sequence was verified using BigDye 3.1 sequencing chemistry and an ABI3130 Genetic Analyzer. The remainder of the culture was supplemented with glycerol to a final concentration of 10%, divided into 25 ml aliquots, and stored at –80°C until needed for expression.

### Expression and purification of recombinant Hb

25 ml of starter culture (above) was added to 1250 ml of TB media containing ampicillin (100 µg/ul) and kanamycin (50 µg/ul) and distributed evenly amongst five 1 L Erlenmeyer flasks. Cultures were grown at 37°C while shaking at 200 rpm until the absorbance at 600 nm reached 0.6–0.8. Hb expression was induced by supplementing the media with 0.2 mM isopropyl β-D-1-thiogalactopyranoside, 50 µg/ml of hemin, and 20 g/L of glucose and the culture was incubated at 28°C for 16 hr while shaking at 200 rpm. Once expression had been completed, dissolved $O_2$ was removed by adding sodium dithionite (1 mg/ml) to the culture, which was promptly saturated with CO for 15 min. Bacterial cells were then pelleted by centrifugation and Hb purified by ion exchange chromatography according to *Natarajan et al., 2011*.

It should be noted that the β82Asn residue of human Hb Providence is relatively uncommon in that it slowly undergoes post-translational deamidation in vivo to form aspartic acid, with the latter residue (β82Asp) comprising ~67–75% in mature mixed blood (*Bardakjian et al., 1985*; *Perutz et al., 1980*). While it is unknown to what degree *H. gigas* β/δ82Asn was catalyzed into Asp in nature, $O_2$ binding data (see below) of this species were collected from freshly purified recombinant samples for which only one peak—presumably β/δ82Asn—was resolved during chromatography (data not shown).

Additionally, this reaction is dependent on the local protein environment (*Robinson, 2002*), specifically two nearby residues β143His and β83Gly (*Perutz et al., 1980*). Importantly, the latter residue was replaced by β/δ83Ser on the Steller's sea cow branch (*Figures 1B and 3B*), which is expected to slow (but not stop) the rate of deamidation (*Robinson, 2002*). Regardless, since the two Hb Providence isoforms have similar $O_2$ affinities and functional properties (*Bardakjian et al., 1985*; *Bonaventura et al., 1976*; *Charache et al., 1977*) it is unlikely that the presence of β/δ82Asp in the Steller's sea cow blood would meaningfully alter the results and interpretations presented herein.

## Functional analyses of Hbs

$O_2$-equilibrium curves for HbA containing solutions (0.25-1.0 mM heme in 0.1 M HEPES/0.0005 M EDTA buffers) were measured at 25 and 37°C using the thin film technique (*Weber, 1992*), while curves for the three pre-natal Hb isoforms (0.25 mM heme in 0.1 M HEPES/0.0005 M EDTA) were measured at 37°C using a multi-cuvette tonometer cell described by *Lilly et al., 2013*. Hb solutions varied in their pH (range: 6.8–7.9), chloride concentration (0 or 0.1 M KCl), and organic phosphate concentration (zero or twofold molar excess of DPG relative to tetrameric Hb concentrations) in order to test the influence of these cofactors on Hb function. Each Hb solution was sequentially equilibrated with gas mixtures of three to five different oxygen tensions ($PO_2$) that result in Hb-$O_2$ saturations between 30 to 70%. Hill plots (log[fractional saturation/[1-fractional saturation]] vs. log$PO_2$) constructed from these measurements were used to determine the $PO_2$ ($P_{50}$) and the cooperativity coefficient ($n_{50}$) at half-saturation, from the $\chi$-intercept and slope of these plots, respectively. By this method, the $r^2$ determination coefficients for the fitted curves exceed 0.995 and the standard errors (SEM) are less than 3% of the $P_{50}$ and $n_{50}$ values (*Weber et al., 2014*). A linear regression was fit to plots of log$P_{50}$ vs. pH, and the resulting equation was used to estimate $P_{50}$, $Cl^-$ effect, and DPG effect values (±SE of the regression estimate) at pH 7.20 for HbA samples, and pH 7.10 for Gower I and HbF samples (to account for the lower pH of pre-natal blood). The slope of these plots (Δlog$P_{50}$/ΔpH) represented the Bohr effect. $P_{50}$ values at 25 and 37°C were used to assess the thermal sensitivity of the Hbs by calculating the apparent enthalpy of oxygenation using the van't Hoff isochore:

$$\Delta H = 2.303\ R \times \Delta \log P_{50} \times (1/T_1 - 1/T_2)^{-1}$$

where R is the universal gas constant and $T_1$ and $T_2$ are the absolute temperatures (°K) at which the $P_{50}$ values were measured. All ΔH values were corrected for the heat of $O_2$ solubilization (12.55 kJ $mol^{-1}$ $O_2$).

## Solubility assay

Ammonium sulfate was added to Hb solutions (0.074±0.004 mM $Hb_4$) to generate final concentrations that ranged from 0 to 3.5 M. These solutions were incubated for 60 min at 37°C and the remaining soluble Hb was measured via Drabkin's reagent, according to the manufacturer's instructions (Sigma-Aldrich).

## Homology modeling

To assess the structural effect of the *H. gigas* specific β/δ replacements on the DPG binding site, homology models of ancestral dugongid and Steller's sea cow Hb were constructed using the SWISS-MODEL server (*Waterhouse et al., 2018*) using the three-dimensional human deoxy structure with DPG bound (PDB: 1B86) as a template (*Richard et al., 1993*). The sequence conservation of amino acid residues implicated in DPG binding was calculated by the ConSurf Server (*Ashkenazy et al., 2016*) from a subsample of 51 mammalian beta-type hemoglobin chains downloaded from GenBank (*Supplementary file 1c*). Homology models were visualized with UCSF Chimera (*Pettersen et al., 2004*). To assess the structural differences between *H. gigas* and *D. dugon* HbF ($α_2γ_2$), homology models of these proteins were created as above, but with human deoxy HbF used as a template (PDB: 4MQJ).

## Acknowledgements

This manuscript is dedicated to the memory of our dear friend Joseph (Joe) Bonaventura for his pioneering work on the human Hb Providence Asn/Asp proteins. We thank Mike Gaudry, Diana

Hanna, and Elin Ellebæk Petersen for technical assistance, Chandrasekhar Natarajan for providing us with a hemoglobin expression plasmid, and Jay Storz for constructive feedback on an earlier version of this manuscript. Authorization to use paintings by Carl Buell was kindly provided by John Gatesy. This study was supported by NSERC (Canada) Discovery and Accelerator Supplement Grants (RGPIN/238838–2011, RGPIN/412336–2011, and RGPIN/06562–2016 to KLC; RGPIN/261924–2013 and RGPIN/446005–2013 to CJB), an NSERC Postgraduate Scholarship (AVS), the Faculty of Science and Technology, Aarhus University (REW), and the Independent Research Fund Denmark (AF; Danmarks Frie Forskningsråd DFF-4181–00094).

## Additional information

### Funding

| Funder | Grant reference number | Author |
|---|---|---|
| Natural Sciences and Engineering Research Council of Canada | RGPIN/238838-2011 | Kevin L Campbell |
| Natural Sciences and Engineering Research Council of Canada | RGPIN/412336-2011 | Kevin L Campbell |
| Natural Sciences and Engineering Research Council of Canada | RGPIN/06562-2016 | Kevin L Campbell |
| Natural Sciences and Engineering Research Council of Canada | RGPIN/261924-2013 | Colin J Brauner |
| Natural Sciences and Engineering Research Council of Canada | RGPIN/446005-2013 | Colin J Brauner |
| Danmarks Frie Forskningsfond | DFF-4181-00094 | Angela Fago |

The funders had no role in study design, data collection and interpretation, or the decision to submit the work for publication.

### Author contributions

Anthony V Signore, Conceptualization, Data curation, Formal analysis, Validation, Investigation, Visualization, Methodology, Writing – original draft, Project administration, Writing – review and editing; Phillip R Morrison, Data curation, Investigation, Methodology, Writing – review and editing; Colin J Brauner, Resources, Supervision, Funding acquisition, Project administration, Writing – review and editing; Angela Fago, Roy E Weber, Resources, Data curation, Formal analysis, Supervision, Funding acquisition, Validation, Project administration, Writing – review and editing; Kevin L Campbell, Conceptualization, Resources, Data curation, Formal analysis, Supervision, Funding acquisition, Validation, Visualization, Writing – original draft, Project administration, Writing – review and editing

### Author ORCIDs

Anthony V Signore ⓘ http://orcid.org/0000-0002-0664-0864
Kevin L Campbell ⓘ http://orcid.org/0000-0001-7005-7086

### Decision letter and Author response

Decision letter https://doi.org/10.7554/eLife.85414.sa1
Author response https://doi.org/10.7554/eLife.85414.sa2

# Additional files

## Supplementary files

• Supplementary file 1. Raw oxygen binding data and GenBank accession numbers. (a) Intrinsic oxygen affinities ($P_{50}$, mmHg) for the adult-expressed hemoglobins ($\alpha_2\beta/\delta_2$) of the Florida manatee (*Trichechus manatus latirostris*), dugong (*Dugong dugon*), Steller's sea cow (*Hydrodamalis gigas*), and the ancestral dugongid ('Anc. dugongid'), and their sensitivity to allosteric effectors at 25 and 37°C in 0.1 M HEPES buffer. All values are corrected to pH 7.2. (b) Intrinsic oxygen affinities ($P_{50}$, mmHg) for the prenatally-expressed hemoglobins Gower I ($\zeta_2\varepsilon_2$) and HbF ($\alpha_2\gamma_2$) of Steller's sea cow (*Hydrodamalis gigas*) and HbF ($\alpha_2\gamma_2$) of the dugong (*Dugong dugon*), and their sensitivity to allosteric effectors at 37°C in 0.1 M HEPES buffer. All values are corrected to pH 7.1 to account for the lower pH of prenatal blood. (c) Accession numbers for beta-globin amino acid sequences used for ConSurf analyses.

• MDAR checklist

## Data availability

Source data files for all data presented are provided as Excel files. Accession numbers for previously published nucleotide sequences used in this study are provided in the manuscript and supporting file.

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
