## [Editor Report]

In this important study, the authors analyze hemoglobin (Hb) from Steller's sea cow [extinct ~250 years ago] and compare it to (sub)tropical sea cows using ancestral sequence reconstruction and site-directed mutagenesis. They convincingly show that Steller's sea cow's Hb had decreased oxygen affinity and increased solubility, indicating adaptation to cold environments. Notably, a single amino acid change accounts for most observed biochemical differences. Additionally, the discovery of a Hb insensitive to DPG adds to the significance of the findings, making this piece an interesting and informative read to all those interested in evolution and adaptation at the molecular level; after all, Hb continues to surprise us.

---

## [Decision Letter]

**Decision letter after peer review:**

Thank you for submitting your article "Evolution of an extreme hemoglobin phenotype contributed to the sub-Arctic specialization of extinct Steller's sea cows" for consideration by *eLife*. Your article has been reviewed by 3 peer reviewers, and the evaluation has been overseen by a Reviewing Editor and Christian Landry as the Senior Editor. The following individual involved in review of your submission has agreed to reveal their identity: Federico Hoffmann (Reviewer #3).

The reviewers have discussed their reviews with one another, and the Reviewing Editor has drafted this to help you prepare a revised submission. Before we dig into specifics, let me add that we all were very pleased with your paper and they thought as highly as I did about this manuscript and had very few issues to note.

Essential revisions:

1) All reviewers saluted the concerns of reviewer #2 regarding maternal/fetal hemoglobins. We understand the Signore et al. (2019) Manatee paper in MBE included information pertaining to the non-adult hemoglobins, and they found no evidence for positive selection in the Steller's Sea Cow. In addition, it also seems based on the alignment there are no Stellers specific mutations; however, it may be worth referencing briefly in the discussion, if not including something in the methods/results using the ~16 Stellers individuals used in the manuscript. A careful reader might raise the same question, especially given the volume devoted to the subject in the discussion.

2) Overall, the manuscript would benefit from a stronger molecular evolution component. Your 2019 MBE paper correctly seems to have already done this, without traces of adaptive evolution. This is pretty remarkable and it may be helpful and important to put your biochemical results in that context, namely, in the light of your MBE conclusions.

*Reviewer #2 (Recommendations for the authors):*

On page 8 in the "Paleophysiology of Steller's sea cows" section, there is a lot of discussion and speculation regarding fetal/embryonic hemoglobin. I wonder what substitutions there are in zeta, γ and epsilon, and where they might fall out on the 3D structure. I am not necessarily suggesting additional functional experiments be performed like was done with the adult Hb, but unless there are any specific barriers to assessing at the sequence level (which were possible for the MBE paper), it would seem enlightening to take a peek at those genes as well.

Figure 2 could use a redo. For panels C-F, I think it would benefit from being a box plot rather than a bar chart. In either case, in panel A, the error bars for the "stripped" cannot be seen as it is masked by the solid black.

Figure 3, the light blue vs dark blue is a bit difficult to infer in the whole-molecule image. It isn't a problem in the zoom-in panel. I'm not sure what other shade to suggest, or perhaps it would be remedied by being larger.

---

## [Author Response]

Essential revisions:1) All reviewers saluted the concerns of reviewer #2 regarding maternal/fetal hemoglobins. We understand the Signore et al. (2019) Manatee paper in MBE included information pertaining to the non-adult hemoglobins, and they found no evidence for positive selection in the Steller's Sea Cow. In addition, it also seems based on the alignment there are no Stellers specific mutations; however, it may be worth referencing briefly in the discussion, if not including something in the methods/results using the ~16 Stellers individuals used in the manuscript. A careful reader might raise the same question, especially given the volume devoted to the subject in the discussion.

We recognize and accept the reviewers’ valid points here. Briefly, it is correct that our previous analysis of the genes encoding the pre-natal hemoglobin (Hb) isoforms of Steller’s sea cows did not show any evidence for positive selection, and the same is also true for the post-natal HBA (α) and HBB/HBD (β/δ) loci of this species. However, the HBG gene (which encodes the γ globin subunit of HbF, which is fetal-expressed in simian primates although it is generally accepted to be solely expressed in the embryonic stage of most other mammalian species) does exhibit four *H. gigas* specific replacements – two of which (γ38Thr→Ile and γ101Glu→Asp) are located within key structural regions of the protein and hence are potentially functionally relevant. Intriguingly, past research on manatee calves suggested that expression of HbF may be sustained throughout fetal development (Farmer et al. 1979). Thus, as part of a related set of experiments in collaboration with Phillip Morrison and Colin Brauner (University of British Columbia), we generated and tested both a known early embryonic Steller’s sea cow Hb isoform (Hb Gower I; ζ2ε2) and the putative fetal isoforms (HbF; α2γ2) of Steller’s sea cows and dugongs to examine this potential further. Notably, these (as of yet unpublished) experiments revealed that the O2 binding properties of sirenian HbF are very similar to the embryonic expressed Hb isoforms of other mammals (including Steller’s sea cow Gower I), though the dugong and Steller’s sea cow HbF proteins revealed discernable differences in O2 affinity and cooperativity that are consistent with the *H. gigas* γ38Thr→Ile and γ101Glu→Asp replacements. Additionally, our (also as of yet unpublished) analysis of the upstream promoter region of sirenian HBG also revealed that the “CACCC” element—which is essential to suppress HBB (β-chain) expression in the fetal stage—is intact in manatees (consistent with the findings of Farmer et al. 1979) but mutated in dugongs and Steller’s sea cows. Taken together, it is thus highly likely that the sole Hb isoform in the fetal circulation of dugonids is HbA (the adult-expressed protein) as is found in the vast majority of mammalian species (hence why we focused on only this specific isoform in the discussion of the initial submission). Based on the consensus of the reviewers, however, we now recognize that this conclusion might not be readily apparent to a general readership lacking this (unpublished) information.

Accordingly, and owing to the inferred biological importance of the described Steller’s sea cow β/δ82Lys→Asn replacement for maternal/fetal gas exchange, we have elected to incorporate a discussion of the physiological and structural characteristics (including one main figure [Figure 4], one supplementary figure [Figure S7], and one supplementary table [Table S2]) of the above three pre-natal Hb isoforms together with an analysis of the upstream HBG promoter regions (Figure S6) into the revised manuscript to fully address the reviewers’ concerns here. Notably, this inclusion necessitated the addition of two new authors on the study (a change approved by all co-authors). We hope this information fully addresses the points raised by reviewer #2 while adding an important new dimension to our understanding of Steller’s sea cow evolution and life history.

M. Farmer, R.E. Weber, J. Bonaventura, R.C. Best, D. Domning, Functional properties of hemoglobin and whole blood in an aquatic mammal, the Amazonian manatee (*Trichechus inunguis*). Comp Biochem Physiol A 62, 231–238 (1979).

2) Overall, the manuscript would benefit from a stronger molecular evolution component. Your 2019 MBE paper correctly seems to have already done this, without traces of adaptive evolution. This is pretty remarkable and it may be helpful and important to put your biochemical results in that context, namely, in the light of your MBE conclusions.

We agree that the lack of statistically significant signatures of positive selection in the Steller’s sea cow globin genes (especially HBB/HBD given the pronounced functional effects of the β/δ82Lys→Asn replacement) is surprising. However, it is worth noting that tests for positive selection of the woolly mammoth HBB/HBD gene were also inconclusive despite obvious functional changes consistent with cold adaptation (Campbell et al. 2010). A more recent study on penguins (Signore et al. 2021) similarly failed to detect positive selection in the α and β genes encoding their hemoglobin, despite clear directional changes in Hb properties (increased O2 affinity and Bohr effect) that accompanied their secondary aquatic adaptation. We thus have included a short discussion in the main text that highlights these exceptions to address this unexpected discord between the molecular and functional evolution of hemoglobin in this extinct marine mammal lineage.

K.L. Campbell, J.E.E. Roberts, L.N. Watson, J. Stetefeld, A.M. Sloan, A.V. Signore, J.W. Howatt, J.R.H. Tame, N. Rohland, T-J. Shen, J.J. Austin, M. Hofreiter, C. Ho, R.E. Weber, A. Cooper. Substitutions in woolly mammoth hemoglobin confer biochemical properties adaptive for cold tolerance. Nature Genetics 42(6):536-540 (2010).

A.V. Signore, M.S. Tift, F.G. Hoffmann, T.L. Schmitt, H. Moriyama H., J.F. Storz, Evolved increases in hemoglobin-oxygen affinity and the Bohr effect coincided with the aquatic specialization of penguins. Proc Natl Acad Sci 118, e2023936118 (2021).

Reviewer #2 (Recommendations for the authors):On page 8 in the "Paleophysiology of Steller's sea cows" section, there is a lot of discussion and speculation regarding fetal/embryonic hemoglobin. I wonder what substitutions there are in zeta, γ and epsilon, and where they might fall out on the 3D structure. I am not necessarily suggesting additional functional experiments be performed like was done with the adult Hb, but unless there are any specific barriers to assessing at the sequence level (which were possible for the MBE paper), it would seem enlightening to take a peek at those genes as well.

We agree. Thus, in addition to the physiological data collected from the three pre-natal isoforms (Figure 4A), we now also include a homology modelled hemoglobin structure of *H. gigas* HbF (Figure S7) that show where the derived Steller’s sea cow replacements are located on the 3D structure, together with structural effects the γ38Thr→Ile and γ101Glu→Asp replacements (Figure 4B-E).

Figure 2 could use a redo. For panels C-F, I think it would benefit from being a box plot rather than a bar chart. In either case, in panel A, the error bars for the "stripped" cannot be seen as it is masked by the solid black.

We appreciate the reviewer’s efforts to improve this figure but, unfortunately, box plots cannot be produced with these particular data. As pH is a potent modulator of Hb–O_2_ affinity, all of the measurements presented in Figure 2 were (by convention) standardized to a fixed pH (in this case pH 7.20). As the titration of each sample/treatment to precisely pH 7.20 is impractical, we instead produce measurements at multiple pHs and use these data to calculate the value at pH 7.20 with a linear equation (± SE of the regression estimate). Therefore, it is not possible to produce box plots of these data as they are not means ± SEM. However, we have modified Figure 2A and 2B to improve visibility of the error bars and made amendments to both the main text and figure caption to clarify the type of data presented.

Figure 3, the light blue vs dark blue is a bit difficult to infer in the whole-molecule image. It isn't a problem in the zoom-in panel. I'm not sure what other shade to suggest, or perhaps it would be remedied by being larger.

We too grappled with trying to optimize this figure prior to submission, with the aim of trying to balance showing all (or at least most) of the Steller’s sea cow specific replacements while also highlighting the DPG binding pocket differences between Steller’s sea cows and their relatives. To remedy this issue, we have thus elected to increase the size of the whole molecule image on Figure 3 and also now include enlarged ‘front’ and ‘back’ images of this protein in the supplementary section so that all changes can be easily discerned (Figure S2).